# A New Digital Method to Quantify the Effects Produced by Carriere Motion Appliance

**DOI:** 10.3390/jpm13050859

**Published:** 2023-05-20

**Authors:** Aldara Rosalía Nercellas Rodríguez, Pedro Colino Gallardo, Álvaro Zubizarreta-Macho, Carlos Colino Paniagua, Alfonso Alvarado Lorenzo, Alberto Albaladejo Martínez

**Affiliations:** 1Department of Surgery, Faculty of Medicine, University of Salamanca, 37008 Salamanca, Spain; aldararosalia.nercellas@usal.es (A.R.N.R.); alfonsoalvaradolorenzo@gmail.com (A.A.L.); albertoalbadalejo@hotmail.com (A.A.M.); 2Department of Orthodontics, European University Miguel de Cervantes, 47012 Valladolid, Spain; pgo.pcolino@odontologiaucam.com (P.C.G.); carloscolinopaniagua@gmail.com (C.C.P.); 3Department of Endodontics, Faculty of Health Sciences, Alfonso X El Sabio University, 28691 Madrid, Spain

**Keywords:** Carriere Distalizer, distalization, derotation, orthodontics, digital

## Abstract

The aim of this study was to analyze a novel digital technique to quantify the distal tooth displacement and derotation angle produced by the Carriere Motion Appliance (CMA). Twenty-one patients with a class II molar and canine relationship underwent orthodontic treatment with CMA. All patients were exposed before (STL1) and after the CMA placement (STL2), submitted to a digital impression, and afterwards, data were uploaded to specific cephalometric software to allow automatic mesh network alignment of the STL digital files. Subsequently, the distal tooth displacement of the upper canines and first upper molars, as well as the derotation angle of the first upper molars were analyzed using the Pearson correlation coefficient (ρ). Repeatability and reproducibility were analyzed using Gage R&R statistical analysis. An increase in canine displacement was correlated with an increase in contralateral canine displacement (ρ = 0.759; *p* < 0.000). An increase in canine displacement was correlated with an increase in molar displacement (ρ = 0.715; *p* < 0.001). An increase in upper first molar displacement was correlated with an increase in the contralateral upper first molar displacement (ρ = 0.609; *p* < 0.003) and the canine displacement (ρ = 0.728; *p* < 0.001). The distal tooth displacement showed a repeatability of 0.62% and reproducibility of 7.49%, and the derotation angle showed a repeatability of 0.30% and reproducibility of 0.12%. The novel digital measurement technique is a reproducible, repeatable, and accurate method for quantifying the distal tooth displacement of the upper canine and first upper molar, as well as the derotation angle of the first upper molars after using CMA.

## 1. Introduction

Class II malocclusion correction using orthodontic approaches is still a challenge for clinicians [1]. Therefore, different treatment approaches for skeletal or dental class II malocclusion have been proposed to stimulate mandible growth using functional orthodontic appliances during peak pubertal growth [1]. However, orthodontic treatment in late adolescence may require fixed orthodontic appliance therapy with intermaxillary elastics or molar distalization devices [1]. Hence, extraoral orthodontic anchorage devices have been suggested for the correction of dental class II malocclusions. However, patient cooperation is required to achieve the desired treatment outcome [2,3]. Therefore, intramaxillary orthodontic appliances, such as repelling magnets, superelastic nickel–titanium archwires, coil springs on a sectional archwire (Distal Jet or Jones Jig), or springs in beta-titanium alloy (Pendulum) have been proposed to solve cooperation drawbacks [2,3,4]. However, they are too complex, too compliance-oriented, prone to breakage [2], and may lead to undesirable effects such as inclination change, specifically regarding mandibular and maxillary incisor inclination [3,5], mesial movement of anchoring teeth [3,6], molar tipping, open bite [5], an increase in the overjet, and an increase in the vertical facial dimension [3]. The Carriere Motion three-dimensional class II appliance, or Carriere Distalizer apparatus (Henry Schein Inc., Melville, New York, NY, USA), is a fixed orthodontic intermaxillary appliance introduced in 2004 for the correction of dental class II malocclusions. The CMA (Henry Schein Inc., NY, USA) is composed of two rigid, nickel-free stainless-steel rods bilaterally attached to the upper canines or premolars (the short model) and the first upper molars. The posterior end of the arm is a ball that articulates on the molar pad, and the canine or premolar pad contains a hook that is used for the placement of intermaxillary elastics [7]. The lower arch serves as the primary source of anchorage for class II corrections, either via a band-anchored lower lingual arch or a removable clear retainer [1]. This mechanism allows the use of intermaxillary elastics from the upper canines or premolars to the lower molars [1]. The fundamental biomechanics described by the author are based on allowing derotation and distalization of the upper first molars and the posterior segments as a unit [7], creating a class I molar and canine relationship in the first phase. Then, the case can be finished with any technique in the posterior dental alignment phase [7]. Growing patients with fully erupted molars are ideal candidates, but adults can also be treated [1,7]. Permanent use of intermaxillary elastics (22 h) is required, and phase I treatment usually takes five to eight months to complete [8]. There is evidence that the CMA (Henry Schein Inc., NY, USA) provides simple [2], efficient, and effective [8] correction of class II malocclusions. Among the main advantages of the CMA (Henry Schein Inc., NY, USA) are its insertion at the beginning of therapy when compliance is still high [7], its easy insertion and removal, its low invasiveness compared with bone-anchored distalization appliances, its elegant design, and the reduced size of the apparatus [9]. In addition, it offers a more positive overall experience and has fewer negative comfort-related side effects compared with other class II correction devices [10]. Although the CMA (Henry Schein Inc., NY, USA) has gained popularity among clinicians over the last decade, there are still few available studies that have evaluated the treatment efficacy and effects produced by this device for class II malocclusion correction [1,8,11]. Existing studies include case reports [12,13,14,15], cohort studies [1,5,8,9,11,16,17], and a systematic review and meta-analysis [18]. Several of these studies concluded that changes are primarily dentoalveolar in nature [1,5,8,11,18,19]. However, these previous studies focused on analyzing the effects of the CMA (Henry Schein Inc., NY, USA) using traditional two-dimensional cephalometric recordings on radiographs [1,5,8,11,17] or with three-dimensional but highly invasive techniques, such as cone-beam computed tomography [16,20,21]. It is also worth noting that rotational changes can be observed using CBCT but not in conventional lateral cephalometric analysis [16], which is considered essential in the correction of dental class II malocclusions and is the main correction mechanism described by the author, who states that the CMA (Henry Schein Inc., NY, USA) produces “a distal rotational movement of the maxillary first molar around their palatal roots” [7]. Maxillary first molars are often rotated with the mesiobuccal cusp displaced in a palatal direction, resulting in a tendency for a class II molar relationship [22]. Therefore, some parameters have been proposed to analyze the position of the maxillary first molar [23,24]. In 1956, Henry et al. measured the angle between the median raphe and a line through the buccal cusps of the molar [23], and Friel et al. evaluated the angle formed between the mesiobuccal and mesiolingual cusps with the middle raphe [24]. However, Ricketts analyzed the line formed between the distobuccal and mesiolingual cusps [25]. Additionally, derotation of the upper molars has become relevant in the present trend of nonextraction treatment, which causes an increase in space [26].

The aim of the present study was to describe a novel digital technique to quantify the distal tooth displacement and derotation angle produced by the CMA (Henry Schein Inc., NY, USA) using a repeatable and reproducible measurement digital technique. The null hypothesis (H_0_) states that the CMA (Henry Schein Inc., NY, USA) does not affect distal tooth displacement and the derotation angle.

## 2. Experimental Section

### 2.1. Study Design

This in vivo study was performed at the Department of Orthodontics of the University of Salamanca (Salamanca, Spain) and the Department of Orthodontics of the European University Miguel de Cervantes (Valladolid, Spain) between November 2020 and March 2021. It was authorized by the Ethical Committee of the Faculty of Medicine and Dentistry, Salamanca University (Salamanca, Spain) in December 2020 (process no. 23/2020). The patients gave their consent to provide the STL digital files before and after CMA placement from the intraoral scan (True Definition, 3M ESPE™, Saint Paul, MN, USA). Name of the Registry: Predictability of Distalization and Derotation of the Carriere Distalizer. A Clinical Study. Clinical Trial Register ID: NCT05094973. Date of Registration: 26 October 2021. URL of trial registry record: https://register.clinicaltrials.gov/prs/app/template/EditProtocol.vm?listmode=Edit&uid=U0005W0Y&ts=3&sid=S000BED7&cx=-yziw16.

### 2.2. Experimental Procedure

Twenty-one patients (8 men and 13 women), aged between 10 and 17 years old (mean age, 13.5 ± 3.5 years) with a class II molar and canine relationship, were consecutively selected and submitted to treatment with the CMA (Henry Schein Inc., NY, USA) for a mean period of 4 months (Figure 1). Orthodontic informed consent was obtained from all patients/parents. The inclusion criteria were as follows: class II mixed dentition patients with fully erupted molars or permanent dentition, no history of previous orthodontic treatment, complete pre- and post-treatment records available (digital study models), a dental class II relationship, and a class I post-treatment occlusal relationship.

All patients were exposed before placing the CMA to a digital impression through an intraoral scan (STL1) (iTero^®^ Element™ 2, Align Technologies, San Jose, CA, USA) by means of 3D in-motion video imaging technology. The images were captured following the manufacturer’s recommendations via first scanning the occlusal plane, followed by the vestibular and palatal faces. Subsequently, the CMA (Henry Schein Inc., NY, USA) was bilaterally cemented on the upper canine and first upper molars by applying a photo-polymerized composite resin cement (Transbond™ XT, 3M ESPE™, Saint Paul, MN, USA) in the center of the buccal surface of the clinical crown, before etching the enamel buccal surface with 37% orthophosphoric acid (OCTACID JUMBO, Laboratorios claraben S.A, Madrid, Spain) for 20 s and photo-polymerized resin adhesive primer application (Unitek Transbond™ XT, 3M ESPE™, Saint Paul, MN, USA) for 20 s. Buccal hooks were bonded to the mandibular first molars, and a clear, invisible 1 mm thick retainer (Dentaflux, Clear. J. Ripoll, S.L, Madrid, Spain), bilaterally trimmed to adapt to the posterior buttons was inserted (Figure 1D–F). Class II elastics were anchored by hooks to the anterior part of both bars and attached to the hooks. Elastic wear consisted of Force 1e elastics (1/4-inch 6 oz) and Force 2e elastics (3/16-inch 8 oz; Henry Schein Orthodontics), which were worn full-time until the end of CMA treatment (Henry Schein Inc., NY, USA. Finally, a postoperative intraoral impression was performed via an intraoral scan (STL2) (True Definition, 3M ESPE™, Saint Paul, MN, USA).

### 2.3. Alignment Procedure

The STL1 (Figure 2A) and STL2 (Figure 2B) digital files were uploaded to specific cephalometric software (Dolphin Imaging, Dolphin Imaging & Management Solutions, Chatsworth, CA, USA). This ensured the alignment of the STL digital files in the automatic mesh network via placing three dots at the same location in both the STL1 and STL2 digital files (one dot in the right central upper incisor, one dot in the right second upper molar, and one dot in the left second upper molar). Then, the alignment rate was manually checked (Figure 2C).

### 2.4. Measurement Procedure

After the STL digital files had been aligned, distal tooth displacement produced by the CMA (Henry Schein Inc., NY, USA) was analyzed for the right and left upper canines and also in the right and left first upper molars. Distal tooth displacement of the upper canines and first upper molars was measured by drawing a line from the canine cusp and mesiobuccal cusp of the upper first molar, perpendicular to the midline in both the STL1 (Figure 3A) and STL2 digital files (Figure 3B). Subsequently, the distance between both lines was measured to obtain the distal tooth displacement produced by the CMA (Henry Schein Inc., NY, USA) at the canines and first upper molars (Figure 3C).

Additionally, the derotation angle produced by the CMA (Henry Schein Inc., NY, USA) was analyzed in the right and left first upper molars by drawing a polygon formed by its cusps in both the STL1 (Figure 4A) and STL2 digital files (Figure 4B). Subsequently, the STL digital files were aligned (Figure 4C), a line crossing the disto-buccal and mesio-palatine cusps to the middle raphe was drawn, and the angle formed by the midline was measured and compared between the STL1 and STL2 digital files to obtain the derotation angle produced by the CMA (Henry Schein Inc., NY, USA) at the first upper molars (Figure 4D,E).

### 2.5. Validation of the Repeatability and Reproducibility of the Technique

The repeatability and reproducibility of this digital measurement technique—which was used to quantify the distal tooth displacement of both the left and right upper canines, the left and right upper first molars, and the derotation angles of both the left and right upper first molars produced by the CMA (Henry Schein Inc., NY, USA)—were validated. To achieve this, the measures were repeated twice by two operators (Operator A and Operator B), and a Gage R&R statistical analysis was performed.

### 2.6. Statistical Analysis

The statistical analysis was conducted using SAS v9.4 (SAS Institute Inc., Cary, NC, USA) and R (R Foundation for Statistical Computing, Vienna, Austria) and descriptively expressed as means and standard deviations (SD). A correlation analysis was conducted on the distal tooth displacement of both the left and right upper canines, the left and right upper first molars, and the derotation angles of both the left and right upper first molars using Pearson correlation coefficients (ρ). The repeatability and reproducibility of this digital measurement method were analyzed via a Gage R&R statistical analysis. The statistical significance was set at *p* < 0.05.

## 3. Results

The means and SD values for the distal tooth displacement of the left and right upper canines (mm), left and right upper first molars (mm), and the derotation angles of the left and right upper molars (°) are displayed in Table 1 and Figure 5 and Figure 6.

A high correlation was shown between the displacement values of the left upper canine and the left upper first molar (*p* = 0.715; *p* < 0.001) (Figure 5A), and a moderate correlation was shown between the displacement values of the left upper first molar and the derotation angle of the left upper first molar (ρ = 0.497; *p* = 0.022) (Figure 5B).

In addition, a strong association was shown between the displacement values of the right upper canine and right upper first molar (ρ = 0.728; *p* < 0.001) (Figure 6).

The means and SD values for the distal tooth displacement of the upper canines (mm) and upper first molars (mm), as well as the derotation angle of the upper first molars (°), are displayed in Table 2 and Figure 7.

A high correlation was shown between the displacement values of the left and right upper canines (*p* = 0.759; *p* < 0.000) (Figure 7A) and the displacement values of the left and right upper molars (*p* = 0.609; *p* < 0.003) (Figure 7B), and a moderate correlation was shown between the rotation values of the left and right upper first molars (ρ = 0.456; *p* = 0.038) (Figure 7C).

This correlation shows that as the canine displacement increased, the contralateral canine displacement also increased (ρ = 0.759; *p* < 0.000), and as the canine displacement increased, the molar displacement also increased (ρ = 0.715; *p* < 0.001), (ρ = 0.728; *p* < 0.001). Moreover, as the upper first molar displacement increased, the contralateral upper first molar displacement increased (ρ = 0.609; *p* < 0.003), and the canine displacement also increased (ρ = 0.715; *p* < 0.001) (ρ = 0.728; *p* < 0.001).

The Gage R&R statistical analysis of the digital measurement technique used to quantify the distal displacement of both the upper canines and upper molars produced by the CMA showed that the variability attributable to the digital measurement technique was 0.62% (among the measures of each operator) and 7.49% (among the measures of the operators) of the total variability among the samples, respectively. The digital measurement technique used to quantify distal tooth displacement is considered repeatable and reproducible because the variability was lower than 10% (Figure 8 and Figure 9).

The Gage R&R statistical analysis of the digital measurement technique was used to quantify the derotation angle of upper molars produced by the CMA. It showed that the variability values attributable to the digital measurement technique were 0.30% (among the measures of each operator) and 0.12% (among the measures of the operators), of the total variability of the samples, respectively. The digital measurement technique used to quantify the derotation angle of upper molars is considered repeatable and reproducible because the variability was under 10% (Figure 10 and Figure 11).

## 4. Discussion

The results presented in this study reject the null hypothesis (H0) that states that the CMA does not affect distal tooth displacement and the derotation angle.

Different treatment approaches have been used for skeletal or dental class II malocclusions to stimulate the growth of the mandible. Functional orthodontic appliances can be used during the pubertal growth period to treat malocclusions produced due to a mandibular growth deficiency or intramaxillary appliances, [1] with a dentoalveolar effect in cases where there is no skeletal problem or when the dentoalveolar peak has already passed. Jamilian et al. analyzed the dentoskeletal effects produced by two types of functional mandibular advancement appliances, including the twin block, which is one of the most popular appliances for the correction of mandibular retrusion [27]. On the other hand, various intramaxillary anchorage appliances for the distalization of the upper molars, such as the Pendulum, Distal Jet, and Jones Jig, have been described and compared as alternatives to traditional headgear [28,29,30]. However, the CMA appears to be more comfortable for the patient, offers a more positive overall experience, and has fewer negative comfort-related side effects compared with other intramaxillary anchorage appliances for class II treatment [10]. Nevertheless, few studies have evaluated the treatment outcomes of using the CMA for the correction of class II malocclusions. Existing studies include case reports [12,13,14,15], cohort studies [1,5,8,9,11,17,19], and a systematic review and meta-analysis [18], and all of them have concluded that the changes are fundamentally of a dentoalveolar nature [1,5,8,11,18,19]. Thus, the present study analyzed dental displacement, regardless of the skeletal malocclusion.

Based on maxillary canine distalization, several studies have analyzed and quantified the effects produced by the CMA (Henry Schein Inc., NY, USA) in class II malocclusions. Areepong et al. analyzed the distalization motion of the upper canines and found a mean displacement of 2.34 mm ± 1.07 mm in skeletal class I and 2.24 mm ± 1.91 mm in skeletal class II patients [16]. Yin et al. found a mean displacement of the upper canines of 3.5 mm ± 1.7 mm, regardless of the skeletal malocclusions [1]. Other recently reported data are displacements of 3.16 ± 1.89 mm [19] and 1.10 ± 0.89 mm [9]. Additionally, this study found a mean distalization displacement of 1.41 mm, which is comparable with the previously described results. The range between the minimum and maximum distalization displacement values of the upper canines was also similar to those shown in previous studies (0.0–4.7 mm) [1,10].

In terms of molar distalization, Yin et al. showed a distalization displacement of the upper molars of 3.7 mm ± 1.7 mm [1]. On the other hand, Areepong et al. differentiated between patients with skeletal class I and class II malocclusions and reported a molar distalization displacement of 1.92 mm in patients with class I malocclusions and a molar distalization displacement of 1.67 mm in patients with class II malocclusions [16], which are similar to the distalization results of 1.67 mm more recently reported by Wilson et al. [19] and of 0.96 ± 0.80 mm reported by Schmid-Herrmann CU et al. [9]. This study found a mean molar distalization displacement of 1.46 mm, but we observed a maximum distalization values comparable to those presented in the first study carried out by Yin et al. [1].

Chiu et al. [28] and Chaques Asensi et al. [31] reported higher mean distalization displacement values using the Pendulum appliance compared with the distalization displacement values found in the present study using the CMA (Henry Schein Inc., NY, USA). Moreover, Fontana et al. [3] and Kinzinger et al. [4] also reported that the Pendulum appliance produces greater molar distalization displacement compared with the CMA (Henry Schein Inc., NY, USA).

According to the molar derotation shown, the designer of the CMA (Henry Schein Inc., NY, USA) revealed that one of the main biomechanical objectives of the device is “distal rotational movement of the maxillary first molars around their palatal roots” as a class II malocclusion correction mechanism. Additionally, the ability of the CMA (Henry Schein Inc., NY, USA) to derotate the upper molars has been previously analyzed. Daniel Areepong et al. argued that it is not possible to measure the rotations of the molars using the previously described methods, so they suggested the analysis of rotating movements with CBCT scans, producing results between 3.588° and 4.648° [16]. A recent similar study carried out by Wilson et al. did not find a significant rotation of the upper molars using the standard CMA, although a significant rotation was found using the short CMA [19]. The purpose of the present study was to measure the degree of rotation using a method that involves less radiation exposure for the patient and, therefore, a minimally invasive technique. The raphe line and median rugae points were traditionally used as reference points for molar derotation, and Van der Linden showed that these anatomical details are sufficiently stable for use as reference structures [32]. The present study suggests a noninvasive digital measurement technique based on the Ricketts measurement technique, which proposes that the line passes through the distobuccal and mesiolingual cusp of the molars [25]. We also suggest using the Friel measurement technique, which analyzes the angle formed within the middle palatine raphe [24]. In this study group, a mean molar rotation angle of 5.75° and a maximum value of 21.5° were found, which agree with the significant upper molar rotations reported by Areepong et al. [16]. Moreover, no study has specifically examined whether distal molar rotations are the reason behind previously observed distal movements using the CMA (Henry Schein Inc., NY, USA) [16]. We found that the rotation and distalization of the molars do not require a statistical correlation; therefore, a greater derotation of the molars will not necessarily result in space gain in the dental arch, as stated by previous studies [23,26,33]. These results agree with those of Dahlquist et al., who reported that the results of the derotation are unpredictable, with respect to space gain and the mesiodistal movement of the mesiobuccal cusp of the molar. Only in some cases is there a considerable space gain and distal movement of the mesiobuccal cusp at the same time as derotation [22].

## 5. Conclusions

The results show that this novel digital measurement technique is a reproducible, repeatable, and accurate method for quantifying the distal tooth displacement of the upper canines and first upper molars and the derotation angles of first upper molars after using the CMA. Additionally, this clinical study demonstrates that, as the upper canine displacement increases, the contralateral upper canine displacement also increases. As the upper first molar displacement increases, the contralateral upper first molar displacement also increases; and as the upper canine displacement increases, the upper molar displacement also increases. No significant correlation was found between molar distalization and molar derotation.

## Figures and Tables

**Figure 1 jpm-13-00859-f001:**
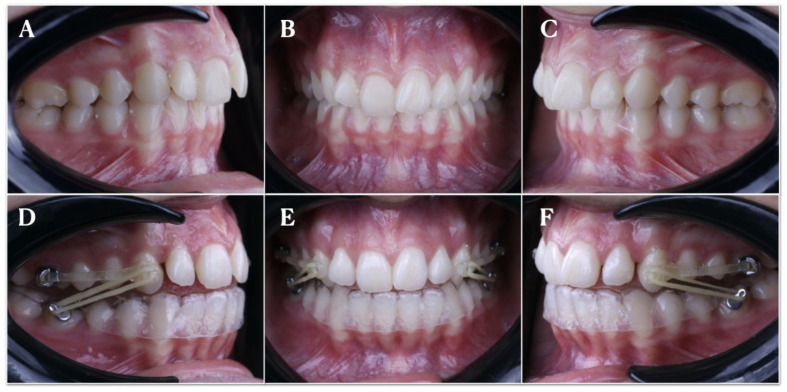
(**A**) Intraoral images of the left lateral, (**B**) frontal, and (**C**) right lateral areas before CMA placement; (**D**) intraoral images of the left lateral, (**E**) frontal, and (**F**) right lateral areas after CMA placement and bilaterally intermaxillary elastics.

**Figure 2 jpm-13-00859-f002:**
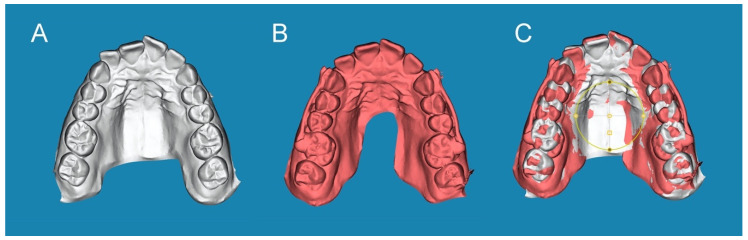
(**A**) STL1 digital file of the upper dental arch before CMA placement. (**B**) STL2 digital file of the upper dental arch after placing the CMA, and (**C**) alignment procedure between the STL1 and STL2 digital files.

**Figure 3 jpm-13-00859-f003:**
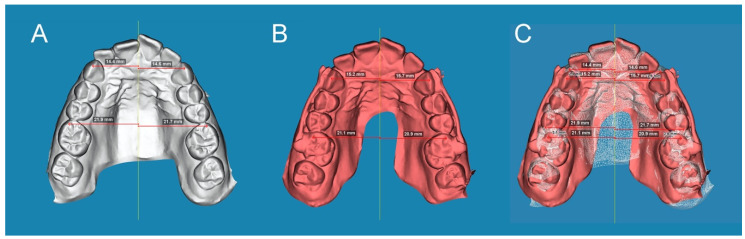
(**A**) STL1 digital file of the upper dental arch with the measures on the canines and first upper molars before CMA placement; (**B**) STL2 digital file of the upper dental arch before placing the CMA with the measures on the canines and first upper molars; and (**C**) alignment procedure between the STL1 and STL2 digital files and digital tooth displacement measurement.

**Figure 4 jpm-13-00859-f004:**
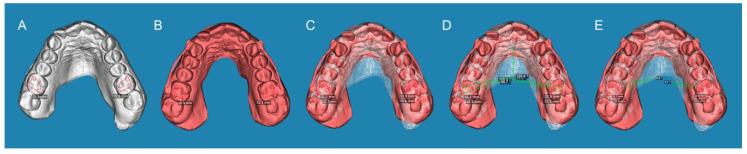
(**A**) STL1 digital file of the upper dental arch before CMA placement with the polygon formed between the cusps of the first upper molars. (**B**) STL2 digital file of the upper dental arch after placing the CMA with the polygon formed between the cusps of the first upper molars. (**C**) alignment procedure for the STL1 and 2 digital files. (**D**) Anterior angles formed by the line that joins the disto-buccal cusp with the mesio-palatine cusp. When cutting this line, the middle raphe and (**E**) derotation angle between the STL1 and STL2 digital files were measured.

**Figure 5 jpm-13-00859-f005:**
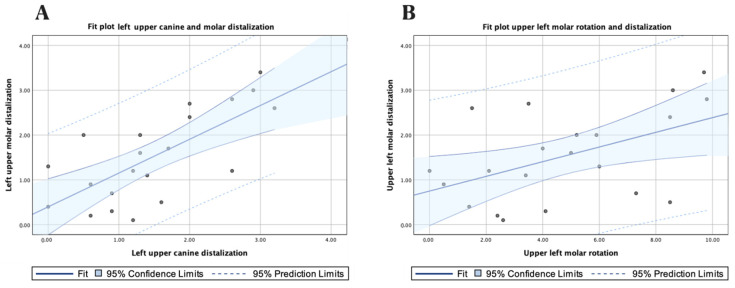
(**A**) Fit plot of the left upper canine and molar distalization; (**B**) fit plot of the left molar rotation and distalization.

**Figure 6 jpm-13-00859-f006:**
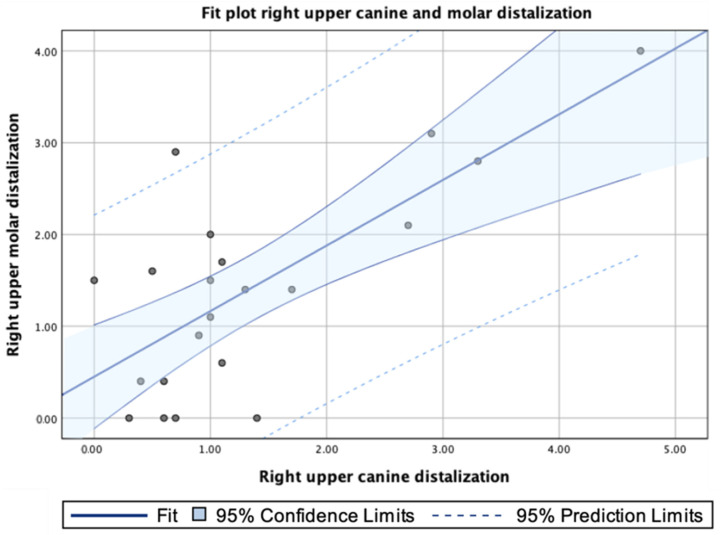
Fit plot of the right upper canine and molar distalization.

**Figure 7 jpm-13-00859-f007:**
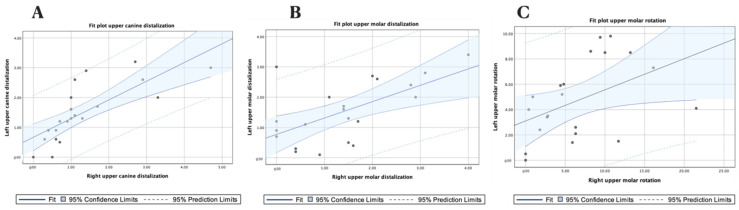
(**A**) Fit plot of the upper canine displacement; (**B**) upper first molar displacement; and (**C**) upper first molar derotation angle.

**Figure 8 jpm-13-00859-f008:**
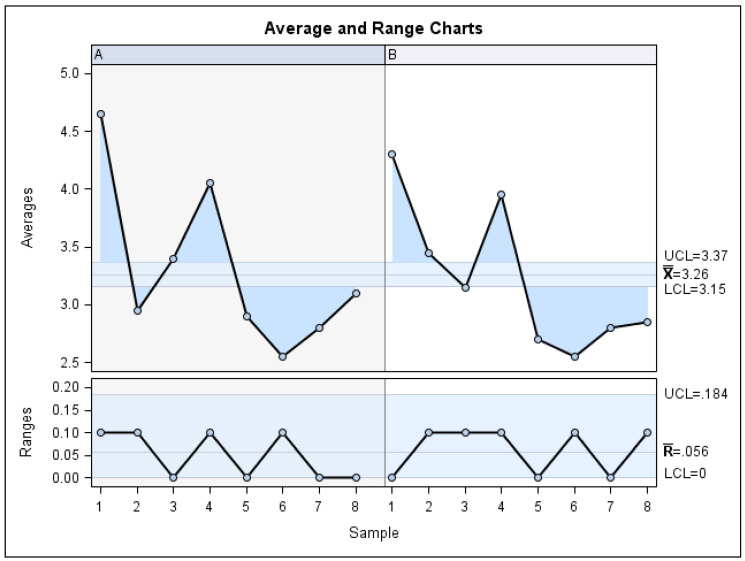
Charts showing the average values of two measures of distal tooth displacement of the upper canines and two measures of the distal tooth displacement of the first upper molars in two patients.

**Figure 9 jpm-13-00859-f009:**
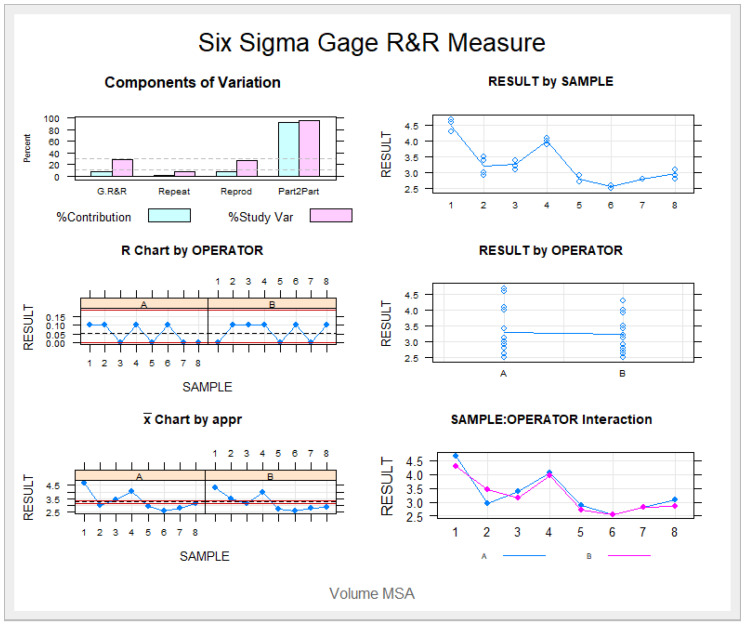
Measurement system analysis related to the distal tooth displacement produced by the CMA with a chart showing the contribution of each component to the total variance (Components of Variation), a mean control chart and a range control chart (R Chart by Operator and x Chart by appr), every measurement point in the graph (Trial by I and Trial by Operator), and the interactions between the operators. (i): Operator interaction.

**Figure 10 jpm-13-00859-f010:**
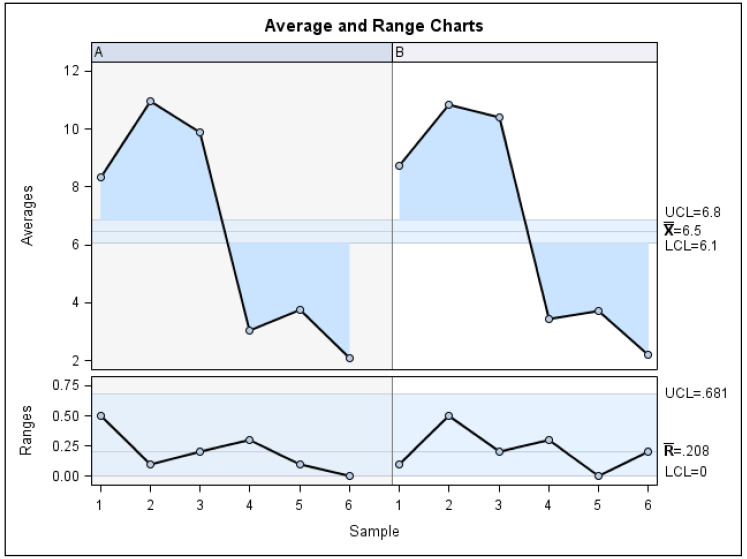
Charts showing the average measures of the derotation angle of the first upper left molar in six patients.

**Figure 11 jpm-13-00859-f011:**
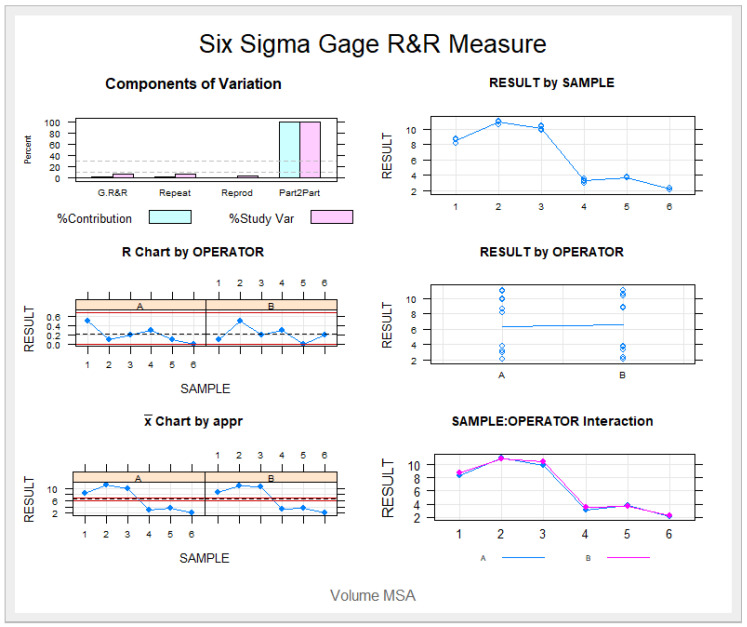
Measurement system analysis related to the derotation angle of upper molars produced by the CMA with a chart of the contribution of each component to the total variance (Components of Variation), a mean control chart and a range control chart (R Chart by Operator and x Chart by appr), every measurement point in the graph (Trial by I and Trial by Operator), and the interactions between the operators. (i): Operator interaction.

**Table 1 jpm-13-00859-t001:** Descriptive statistics of the distal tooth displacement of the left and right upper canines (mm), left and right upper molars (mm), and the derotation angles of the left and right upper molars (°).

	*n*	Mean	SD	Minimum	Maximum
Left upper canine displacement	21	1.500	0.953	0	3.4
Right upper canine displacement	21	1.329	1.152	0	4.7
Left upper molar displacement	21	1.529	1.006	0.1	3.4
Right upper molar displacement	21	1.400	1.133	0	4
Left upper molar derotation angle	21	4.762	3.054	0	9.8
Right upper molar derotation angle	21	6.748	5.673	0	21.5

SD: standard deviation.

**Table 2 jpm-13-00859-t002:** Descriptive statistics of the distal tooth displacement of the upper canines (mm), upper first molars (mm), and the derotation angle of the upper first molars (°).

	*n*	Mean	SD	Minimum	Maximum
Upper canine displacement	42	1.414	1.048	0	4.7
Upper molar displacement	42	1.464	1.060	0	4
Upper molar derotation angle	42	5.755	4.611	0	21.5

SD: standard deviation.

## Data Availability

Data available on request due to restrictions, e.g., privacy or ethical.

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
