# Peer review of "A New Digital Method to Quantify the Effects Produced by Carriere Motion Appliance"

_jpm, 2023, doi:10.3390/jpm13050859_

Round 1

Reviewer 1 Report

Although this study is interesting, there are some flaws in the study design.

The title is too long

The method and material are confusing. Due to the following sentences:

In the last paragraph, the authors say that “This in vitro study was performed at the Department of Orthodontics…..”

On the other hand, at the beginning of the material and method, the authors say that “Twenty-one patients (8 men and 13 women) with Class II molar…”

The 2 above statements from the authors are very confusing. Is this in vitro or in vivo study?

There are some grammatical errors: for example, the word “woman “in the following sentence on page 3 must be replaced by women

On page 3 “Twenty-one patients (8 men and 13 woman) with Class II molar and….”

Authors must explain how this appliance can correct class 2 molar relationship in non-growing adult patients especially with 19 years old. The mean age of patients must be added to the material and methods.

Cephalometric analysis, including the following landmarks, must be added in a separate table to define the severity of skeletal problems.

SNA(°), SNB(°), ANB(°), Wits appraisal (mm), Go-GN to SN plane (°), Y-axis (°), ANS-PNS (mm), Overjet (mm), Overbite (mm), Nasolabial angle (°), IMPA (°)

For expanding the discussion this appliance must be compared with the 2 appliances in the following article and the following article must be cited.

Treatment effects of the R-appliance and twin block in Class II division 1 malocclusion.

Eur J Orthod. 2011 Aug;33(4):354-8. doi: 10.1093/ejo/cjq082.

Minor editing of English language is  required

Author Response

Dear Reviewer 1:

I’m pleased to resubmit the manuscript of the work entitled, “A Repeatable and Reproducible Digital Method to Quantify the Distal Tooth Displacement and Derotation Angle Produced by Carriere Motion Appliance

Reviewer 1: Minor editing of English language required

Response: In order to adapt to the reviewer's 1 comments, we have send the manuscript to the English Editing Service of MDPI. We attached the Certificate.

Reviewer 1: The title is too long

Response: In order to adapt to the reviewer's 1 comments, we have rewritten the title with less words: “A New Digital Method to Quantify the Effects Produced by Carriere Motion Appliance”.

Reviewer 1: The method and material are confusing. Due to the following sentences:

In the last paragraph, the authors say that “This in vitro study was performed at the Department of Orthodontics...”

On the other hand, at the beginning of the material and method, the authors say that “Twenty-one patients (8 men and 13 women) with Class II molar…”

The 2 above statements from the authors are very confusing. Is this in vitro or in vivo study?

Response: In order to adapt to the reviewer's 1 comments, we clarify that this is an in vivo study registered in ClinicalTrials.org. In addition, we have changed the sentence “This in vitro study was conducted in the Department of Orthodontics...” to “This in vivo study was conducted in the Department of Orthodontics...”

Reviewer 1: There are some grammatical errors: for example, the word “woman “in the following sentence on page 3 must be replaced by women

On page 3 “Twenty-one patients (8 men and 13 woman) with Class II molar and...”

Response: In order to adapt to the reviewer's 1 comments, we have replaced the word “woman” by women at the “Materials and Methods” section. Furthermore, we have sent the manuscript to the English Editing Service of MDPI. We attached the Certificate.

Reviewer 1: Authors must explain how this appliance can correct class 2 molar relationship in non-growing adult patients especially with 19 years old. The mean age of patients must be added to the material and methods.

Response: In order to adapt to the reviewer's 1 comments, we have revised the age of the recruited patients of the study and we must apologize because we made a mistake. The older patient selected was 17 years old; therefore, we changed the aged of the patients included in this study, that are all growing patients between 10 and 17 years old. In addition, we have also added the mean age of the patients: 13.5 ± 3.5.

Reviewer 1: Cephalometric analysis, including the following landmarks, must be added in a separate table to define the severity of skeletal problems.

SNA(°), SNB(°), ANB(°), Wits appraisal (mm), Go-GN to SN plane (°), Y-axis (°), ANS-PNS (mm), Overjet (mm), Overbite (mm), Nasolabial angle (°), IMPA (°)

Response: In order to adapt to the reviewer's 1 comments, we claify that the studies that have evaluate the effects produced by the Carriere Motion Appliance concluded that the changes are fundamentally dentoalveolar (Yin K, Han E, Guo J, Yasumura T, Grauer D, Sameshima G. Evaluating the treatment effectiveness and efficiency of Carriere Distalizer: a cephalometric and study model comparison of Class II appliances. Prog Orthod. 2019 Jun 18;20(1):24. doi: 10.1186/s40510-019-0280-2. Kim-Berman H, McNamara JA Jr, Lints JP, McMullen C, Franchi L. Treatment effects of the Carriere® Motion 3D™ appliance for the correction of Class II malocclusion in adolescents. Angle Orthod. 2019 Nov;89(6):839-846. doi: 10.2319/121418-872.1 Wilson B, Konstantoni N, Kim KB, Foley P, Ueno H. Three-dimensional cone-beam computed tomography comparison of shorty and standard Class II Carriere Motion appliance. Angle Orthod. 2021 Jul 1;91(4):423-432. doi: 10.2319/041320-295.1.) Furthermore, due to the short duration (4 months) and the fact that stable structures of the upper arch were superimposed (palatal ridges), effect of growth on STL1 and STL2 changes was considered minimal.

Reviewer 1: For expanding the discussion this appliance must be compared with the 2 appliances in the following article and the following article must be cited.

Treatment effects of the R-appliance and twin block in Class II division 1 malocclusion.

Eur J Orthod. 2011 Aug;33(4):354-8. doi: 10.1093/ejo/cjq082.

Response: In order to adapt to the reviewer's 1 comments, we have added a sentences with the reference suggested by the Reviewer 1 in the “Discussion” section: “Different treatment approaches have been used for skeletal or dental class II malocclusion to stimulate growth of the mandible with functional orthodontic appliances during the pubertal growth period in malocclusions due to mandibular growth deficiency or intramaxillary appliances [Yin K, Han E, Guo J, Yasumura T, Grauer D, Sameshima G. Evaluating the treatment effectiveness and efficiency of Carriere Distalizer: a cephalometric and study model comparison of Class II appliances. Prog Orthod. 2019, 20, 24.] with a dentoalveolar effect in cases where there is no skeletal problem or the dentoalveolar peak has already passed. Jamilian et al, analyzed the dentoskeletal effects produced by two types of functional mandibular advancement appliances, including twin- block, which is one of the most popular appliances for the correction of the mandibular retrusion [Jamilian A, Showkatbakhsh R, Amiri SS. Treatment effects of the R-appliance and twin block in Class II division 1 malocclusion. Eur J Orthod. 2011 Aug;33(4):354-8. doi: 10.1093/ejo/cjq082.]. On the other hand, ”.

We take this opportunity to thank the recommendations and suggestions made by the reviewers to improve the document.

Yours sincerely,

Reviewer 2 Report

I want to thank the authors for this article and the great effort into it.

I have some minor revisions to propose to you to improve your work. Please refer to the following comments:

1. Introduction and Discussion

-please check the way how the references are cited in the text.

2. Conclusions

-I would recommend to re-arrange or rephrasing the conclusions.

3. References

-I would recommend adding more updated references

Author Response

Dear Reviewer 2:

I’m pleased to resubmit the manuscript of the work entitled, “A Repeatable and Reproducible Digital Method to Quantify the Distal Tooth Displacement and Derotation Angle Produced by Carriere Motion Appliance

Reviewer 2: I have some minor revisions to propose to you to improve your work. Please refer to the following comments:

  1. Introduction and Discussion

-please check the way how the references are cited in the text.

Response: In order to adapt to the reviewer's 2 comments, we have checked the references cited in the text.

Reviewer 2: 2. Conclusions

-I would recommend to re-arrange or rephrasing the conclusions.

Response: In order to adapt to the reviewer's 2 comments, we have re-arrange the “Conclusion” section: The results showed that the novel digital measurement technique is a reproducible, repeatable and accurate method for quantifying the distal tooth displacement of upper canine and first upper molar and the derotation angle of first upper molars after using CMA. Additionally, this clinical study demonstrates that the more the upper canine displacement is increased, the more the contralateral upper canine displacement increases, the more the upper first molar displacement is increased, the more the contralateral upper first molar displacement increases, the more the upper canine displacement is increased, the more the upper molar displacement increases, and no significant correlation was found between molar distalization and molar derotation.

Reviewer 2: 3. References

-I would recommend adding more updated references

Response: In order to adapt to the reviewer's 2 comments, we have added content in the main text and updated references and incorporated the following recent and interesting references in the "Introduction" and "Discussion" section:

Schmid-Herrmann CU, Delfs J, Mahaini . Retrospective investigation of the 3D effects of the Carriere Motion 3D appliance using model and cephalometric superimposition. Clin Oral Investig. 2023;27(2):631-643.

Hamilton CF, Saltaji H, Preston CB, Flores-Mir C, Tabbaa S. Adolescent patients' experience with the Carriere distalizer appliance. Eur J Paediatr Dent. 2013, 14, 219-224.

McNamara JA Jr, Franchi L, McClatchey LM, Kim-Berman H. Evaluating new approaches to the treatment of Class II and Class III malocclusions: the Carriere Motion appliance. In: Shroff B, ed. Embracing Novel Technologies in Dentistry and Orthodontics. Monograph 57, Craniofacial Growth Series, Department of Orthodontics and Pediatric Dentistry and Center for Human Growth and Development. Ann Arbor: The University of Michigan; 2021.

Rodrigues H. Unilateral application of the Carriere distalizer. J Clin Orthod. 2011, 45, 177–180.

Rodrigues H. Nonextractin treatment of a Class II Open Bite in an Adult Patient. J Clin Orthod. 2012, 46, 367-371

Singh D. Intraoral Approaches for Maxillary Molar Distalization: Case Series. J Clin Diagnos Res. 2017, 11, ZR01-ZR04.

Pardo Lopez B, DeCaros Villafranca F, Cobo Plana J. Distalizer treatment of an adult Class II Division 2 malocclu- sion. J Clin Orthod. 2006, 40, 561–565.

Areepong, D., Kim, K.B., Oliver, D.R., Ueno, H. (2020) The Class II Carriere Motion appliance. Angle Orthod., 90, 491-499.

Wilson B, Konstantoni N, Kim KB, Foley P, Ueno H. Three-dimensional cone-beam computed tomography comparison of shorty and standard Class II Carriere Motion appliance. Angle Orthod. 2021;91(4):423-432.

Luca L, Francesca C, Daniela G, Alfredo SG, Giuseppe S. Cephalometric analysis of dental and skeletal effects of Carriere Motion 3D appliance for Class II malocclusion. Am J Orthod Dentofacial Orthop. 2022;161(5):659-665.

Barakat D, Bakdach WMM, Youssef M. Treatment effects of Carriere Motion Appliance on patients with class II malocclusion: A systematic review and meta-analysis. Int Orthod. 2021;19(3):353-364. doi:10.1016/j.ortho.2021.05.005

Fouda AS, Attia KH, Abouelezz AM, El-Ghafour MA, Aboulfotouh MH. Anchorage control using miniscrews in comparison to Essix appliance in treatment of postpubertal patients with Class II malocclusion using Carrière Motion Appliance. Angle Orthod. 2022;92(1):45-54.

Hashem AS. Effect of second molar eruption on efficiency of maxillary first molar distalization using Carriere distalizer appliance. Dental Press J Orthod. 2021;26(4):e2119146.

We take this opportunity to thank the recommendations and suggestions made by the reviewers to improve the document.

Yours sincerely,

Reviewer 3 Report

This study analyzed a novel digital technique to quantify the distal tooth displacement and derotation angle produced by Carriere Motion (CM) appliances using a repeatable and reproducible digital measurement technique on 21 patients. In detail, patients were exposed preoperatively (STL1) and postoperatively (STL2) to a digital impression and then using cephalometric software to allow the automatic mesh networks alignment of the STL digital files. The distal tooth displacement of the upper canines and first upper molars and the first upper molars' derotation angle were measured using the Pearson correlation. The repeatability and reproducibility were analyzed by Gage R&R statistical analysis.

The results show that the novel digital measurement technique is a reproducible, repeatable, and accurate method for quantifying the distal tooth displacement of the upper canine and first upper molar and the derotation angle of the first upper molars after using the CM appliance.

The study was well-designed and analyzed. The scientific writing was sound. I would like to recommend it for publication in its present form.

Author Response

Dear Reviewer 3:

I’m pleased to resubmit the manuscript of the work entitled, “A Repeatable and Reproducible Digital Method to Quantify the Distal Tooth Displacement and Derotation Angle Produced by Carriere Motion Appliance

We take this opportunity to thank the recommendations and suggestions made by the reviewers to improve the document.

Yours sincerely,

Round 2

Reviewer 1 Report

Article is publishable.